

**Large contribution of soil emissions to the atmospheric nitrogen**
**budget and their impacts on air quality and temperature rise in**
**North China**
*Tong Sha[1][*], Siyu Yang[1], Qingcai Chen[1], Liangqing Li[1], Xiaoyan Ma[2], Yan-Lin Zhang[3,4],*
*Zhaozhong Feng[3], K. Folkert Boersma[5,6], Jun Wang[7][*]*
[1] School of Environmental Science and Engineering, Shaanxi University of Science and
Technology, Xi'an 710021, China
[2] Key Laboratory for Aerosol-Cloud-Precipitation of China Meteorological
Administration, Nanjing University of Information Science & Technology, Nanjing
210044, China
[3] School of Ecology and Applied Meteorology, Nanjing University of Information
Science & Technology, Nanjing 210044, China
[4] Atmospheric Environment Center, Joint Laboratory for International Cooperation on
Climate and Environmental Change, Ministry of Education (ILCEC), Nanjing
University of Information Science & Technology, Nanjing 210044, China
[5] Satellite Observations Department, Royal Netherlands Meteorological Institute, De
Bilt 3731GA, the Netherlands
[6] Meteorology and Air Quality Group, Wageningen University Wageningen 6708PB,
the Netherlands
[7] Department of Chemical and Biochemical Engineering, Center for Global and
Regional Environmental Research, and Iowa Technology Institute, University of Iowa,
Iowa City, IA, 52242, USA



*Corresponding authors:
Tong Sha: tong-sha@sust.edu.cn
Jun Wang, jun-wang-1@uiowa.edu





**Abstract**

Soil emissions of nitrogen compounds, including NO and HONO, play a
significant role in atmospheric nitrogen budget. However, HONO has been overlooked
in previous research on soil reactive nitrogen (Nr) emissions and their impacts on air
quality in China. This study estimates both soil $NO_x$ and HONO emissions ($SNO_x$ and
SHONO) in North China with an updated soil Nr emissions scheme in a chemical
transport model, the Unified Inputs for WRF-Chem (UI-WRF-Chem). The effects of
soil Nr emissions on $O_3$ pollution, air quality and temperature rise are also studied, with
a focus on two key regions, Beijing-Tianjin-Hebei (BTH) and Fenwei Plain (FWP),
known for high soil Nr and anthropogenic emissions. We find that the flux of $SNO_x$ is
nearly doubled those of SHONO; the monthly contributions of $SNO_x$ and SHONO
account for 37.3% and 13.5% of anthropogenic $NO_x$ emissions in the BTH, and 29.2%
and 19.2% in the FWP during July 2018, respectively. Soil Nr emissions have a
significant impact on surface $O_3$ and nitrate, exceeding $SNO_x$ or SHONO effects alone.
On average, soil Nr emissions increase MDA8 $O_3$ by 16.9% and nitrate concentrations
by 42.4% in the BTH, 17.2% for MDA8 $O_3$ and 42.7% for nitrate in the FWP. Reducing
anthropogenic $NO_x$ emissions leads to a more substantial suppressive effect of soil Nr
emissions on $O_3$ mitigation, particularly in BTH. Soil Nr emissions, via their role as
precursors for secondary inorganic aerosols, can result in a slower increase rate of
surface air temperature. This study suggests that mitigating $O_3$ pollution and addressing
climate change in China should consider the role of soil Nr emission, and their regional
differences.



## 1. Introduction

Surface ozone ($O_3$) is a major air pollutant harmful to human health, terrestrial vegetation, and crop growth (Feng et al., 2022b; Turner et al., 2016; Unger et al., 2020; Yue et al., 2017). China is confronting serious $O_3$ pollution, with the surface $O_3$ concentrations routinely exceeding air quality standards (Li et al., 2019). Although the Chinese Action Plan on Air Pollution Prevention and Control implemented in 2013 has significantly reduced the nationwide anthropogenic emissions of primary pollutants including particulate matter (PM) and nitrogen oxides ($NO_x = NO + NO_2$), the summertime $O_3$ concentrations observed by national ground sites and satellite observations both show an increasing trend of 1-3 ppbv $a^{-1}$ in megacity clusters of eastern China from 2013 to 2019 (Wang et al., 2022b; Wei et al., 2022). Many studies have explored the causes of $O_3$ pollution from the perspective of changes in meteorology and anthropogenic emissions, and attributed the $O_3$ increase to decreased PM levels and anthropogenic $NO_x$ emissions, and adverse meteorological conditions (Li et al., 2021b; Li et al., 2019; Li et al., 2020; Liu and Wang, 2020a, b; Lu et al., 2019).

Soil emissions are an important natural source of reactive nitrogen species, including $N_2O$, $NO_x$, HONO and $NH_3$, and can strongly affect the atmospheric chemistry, air pollution and climate change (Elshorbany et al., 2012; Pinder et al., 2012). It has been acknowledged that the soils emissions account for 12-20% of total emissions of $NO_x$ in global average (Vinken et al., 2014; Yan et al., 2005), and 40-51% in agricultural regions during periods in which fertilizers are applied to soils, resulting in a significant increase in $O_3$ and $NO_2$ concentrations in US (Almaraz, 2018; Romer et





al., 2018; Sha et al., 2021; Wang et al., 2021a), Europe (Skiba et al., 2020) and sub-
Saharan Africa (Huang et al., 2018).
China has a large area of cultivated land (~1.276 ×10$^6$ km$^2$,
http://gi.mnr.gov.cn/202304/t20230414_2781724.html, last access: 18$^{th}$ December
2023), which contributes to one-third of the global nitrogen fertilizer use and has
extensive nitrogen deposition (Liu et al., 2013; Lu and Tian, 2017; Reay, 2008). So far,
only a limited studies focused on the impact of soil NO$_x$ emissions (denoted as SNO$_x$)
on O$_3$ pollution in China (Huang et al., 2023; Lu et al., 2021; Shen et al., 2023; Wang
et al., 2008; Wang et al., 2023a; Wang et al., 2022a). Lu et al. (2021) demonstrated that
the presence of SNO$_x$ in the North China Plain significantly reduced the sensitivity of
surface O$_3$ to anthropogenic emissions. Huang et al. (2023) suggested that substantial
SNO$_x$ could increase the maximum daily 8 h (MDA8) O$_3$ concentrations by 8.0–12.5
μg m$^{-3}$ on average for June 2018 in China. These studies focused only on NO$_x$ emitted
from soils, and neglected that similar soil microbial activities also emit nitrous acid
(HONO). The measurements in laboratory showed that the emission rates of soil HONO
were comparable to those of NO (Oswald et al., 2013; Weber B, 2015). The photolysis
of HONO has been identified to be an important source of atmospheric hydroxyl radical
(·OH), which enhances concentrations of hydroperoxyl (HO$_2$) and organic peroxy
radicals (RO$_2$), accelerating the conversion of NO to NO$_2$, resulting in more
concentrations of O$_3$ and secondary pollutants. Although the sources and formation
mechanisms of HONO are still not fully understood, recent model studies suggested
that HONO emission from soils in the agriculture-intensive North China Plain could



increase the regionally averaged daytime ·OH, O₃, and daily fine particulate nitrate
concentrations (Feng et al., 2022a; Wang et al., 2021b).
Only a few studies simultaneously considered the impact of soil HONO emissions
(denoted as SHONO) along with SNO$_x$ on O₃ and other secondary pollutants (Tan et
al., 2023; Wang et al., 2023b). Wang et al. (2023b) found that the NO$_x$ and HONO
emissions from natural soils (i.e., soil background emissions) increased daily average
O₃ concentrations by 2.0% in Northeast Plain during August 2016 without considering
the contribution from fertilized croplands. Tan et al. (2023) believed that the
contribution of soil NO$_x$ and HONO to O₃ pollution has been in an increasing trend
from 2013 (5.0 pptv) to 2019 (8.0 pptv) in the summer season over the North China
Plain by using the GEOS-Chem model; however the coarse resolution of GEOS-Chem
simulation may not insufficient to resolve the spatial heterogeneity in soil emission
distribution (Lu et al., 2021). Associated with the decreasing anthropogenic emissions
is the increasing contribution of soil emissions to the atmospheric nitrogen budget in
China. Therefore, it is critical to quantify the impact of soil reactive nitrogen (Nr: NO$_x$
and HONO) emissions on O₃ and secondary pollutants.
In this study, we improve the soil Nr emissions scheme in the Unified Inputs
(initial and boundary conditions) for Weather Research and Forecasting model coupled
with Chemistry (UI-WRF-Chem) by considering all potential sources of HONO
published in the literature. Since serious O₃ pollution and high soil emissions always
occurred in summer, a series of sensitivity experiments are conducted to quantify the
coupled and separate impact of SNO$_x$ and SHONO on O₃ and secondary pollutants



during July over the North China, focusing on two city clusters, the Beijing-Tianjin-
Hebei (BTH) region and Fenwei Plain (FWP) region, both of which have the vast areas
of croplands and dense populations and experiencing severe $O_3$ and $PM_{2.5}$ pollutions.
In addition, by quantitatively analyzing the difference in the response of surface $O_3$
concentrations and surface air temperature to the anthropogenic $NO_x$ emissions
reductions in the presence vs. absence of soil Nr emissions, the roles of soil Nr
emissions on $O_3$ mitigation strategies and climate change are also studied. Our study is
designed to address the underestimated role of soil Nr emission in $O_3$ pollution, thereby
providing the scientific basis for $O_3$ mitigation strategies and climate change.
**2. Methodology**
**2.1 Model description**
**2.1.1 Model configurations, input data, and non-soil HONO emission**

The UI-WRF-Chem model, developed upon the standard version of WRF-Chem

3.8.1 (Grell et al., 2005), was used in this study. The 0.625°×0.5° Modern-Era
Retrospective analysis for Research and Applications, Version 2 (MERRA-2) reanalysis
data provide both the meteorological and chemical boundary and initial conditions
(Gelaro et al., 2017). The 0.25° × 0.25° Global Land Data Assimilation System
(GLDAS) data provides the initial and boundary conditions of soil properties, i.e., soil
moisture and temperature (Rodell, 2004). Details of Unified Inputs of meteorological
and chemical position data for UI-WRF-Chem, can be found in recent publications (Li
et al., 2024; Wang et al., 2023c). Anthropogenic emissions are imported from the Multi-
resolution Emission Inventory for China (MEIC: http://www.meicmodel.org/) with a



spatial resolution of 0.25° × 0.25° for the year 2017. Biomass burning emissions are
from the Fire Inventory from NCAR version (FINN, version 1.5,
https://www.acom.ucar.edu/Data/fire/). Biogenic emissions are calculated using the
Model of Emissions of Gases and Aerosols from Nature (MEGAN) version 2.1
(Guenther et al., 2012).
The physical and chemical schemes include the Morrison 2-moment
microphysical scheme (Morrison et al., 2009), Grell 3-D cumulus scheme (Grell and
Dévényi, 2002), RRTMG for both longwave and shortwave radiation scheme (Iacono
et al., 2008), Yonsei University planetary boundary layer scheme (Hong, 2006), Noah
land surface model (Tewari, 2004), and the Carbon Bond Mechanism (CBMZ) for gas-
phase chemistry and the Model for Simulating Aerosol Interactions and Chemistry
(MOSAIC) aerosol module with four sectional aerosol bins and aqueous reactions
(Zaveri et al., 2008; Zaveri and Peters, 1999) are adopted in the UI-WRF-Chem model.
Two nested domains are used, domain one covers China with a horizontal resolution of
27 km and contains 112×112 grid cells, and domain two covers central and eastern
China and its surrounding area with a horizontal resolution of 9 km, containing
196×166 grid cells (study region are shown in Figure S1), both domains have 74 vertical
levels from surface to 50 hPa and 4 levels of soil. The simulations are conducted from
29[th] June to 31[th] July in 2018 with the first 2 days as the spin-up period. The model
outputs from 1[th] to 31[th] July in 2018 are analyzed.
The default WRF-Chem model only considers the gas-phase formation of HONO
(NO + OH → HONO), thus underestimating the HONO concentrations. In this study, in



addition to considering SHONO (details in Section 2.1.2), potential sources of HONO
recognized in recent studies are also taken into account in the current model (Fu et al.,
2019; Li et al., 2010; Ye et al., 2016; Ye et al., 2017; Zhang et al., 2022b; Zhang et al.,
2022a; Zhang et al., 2016; Zhang et al., 2021; Zhang et al., 2020), including traffic
emissions, $NO_2$ heterogeneous reactions on ground and aerosol surfaces, and inorganic
nitrate photolysis in the atmosphere. Through a series of tests and comparisons with
observed surface HONO concentrations, the specific parameterization schemes of
HONO sources adopted in this study are shown in Text S1.
**2.1.2 Parameterization of soil Nr emissions**
The soil Nr emissions schemes in the UI-WRF-Chem model are updated in this
study. The default $SNO_x$ scheme in UI-WRF-Chem, MEGAN v2.1, is replaced by the
Berkeley−Dalhousie−Iowa  Soil  NO  Parameterization  (BDISNP),  and  the
implementation of BDISNP can be found in Sha et al. (2021). Considering that the
baseline year of N fertilizer data is 2006, and the amount of N fertilizer application in
China has changed in the past ten years, we update the N fertilizer data to the year 2018
based on the N fertilizer application data at the province level from the statistical
yearbook (Table S1).
The process of soil HONO emission is similar to that of $NO_x$, as both are
influenced by the physical and chemical characteristics of soils. Consequently, soil
emissions of HONO with consideration of their dependence on land type, soil humidity,
and temperature are also parameterized into the UI-WRF-Chem model. We first map
the soil types measured in Oswald et al. (2013) (collected from 17 ecosystems in Table



S2) into the most closely matching MODIS land cover types in the model following
Feng et al. (2022a), described in Table S3. The optimal emission flux for each MODIS
land cover type is calculated as the average of the measured fluxes from the
category/categories in Oswald et al. (2013) that is/are been mapped into a specific
MODIS classification. We also collect the SHONO data from various ecosystems in
China published in different studies to correct the optimal SHONO fluxes in the model
(Table S4). These ecosystems include semi-arid, fertilized and irrigated farmland in
China. Consequently, the parameterization scheme takes into account the effect of
fertilizer application on the SHONO. After that, the optimal fluxes over the domains
are digested into the model and further scaled online according to the soil temperature
and water content in each model grid at each time step throughout the simulation period
by the following of equation from (Zhang et al., 2016):
$$F_N(\text{HONO}) = F_{N,opt}(\text{HONO}) \cdot f(T) \cdot f(SWC)$$
where $F_{N,opt}(\text{HONO})$ is the optimum flux of SHONO in terms of nitrogen. $f(T)$ and
$f(SWC)$ are the scaling factors of soil temperature ($T$) and water content ($SWC$).
$$f(T) = e^{\frac{Ea}{R}(\frac{T}{T_{opt}} - \frac{1}{T})}$$
$E_a$ is the activation energy of HONO (80 kJ mol⁻¹), R is the gas constant, $T_{opt}$ is
the temperature at which optimum flux is emitted (298.15 K), $T$ is the soil temperature
calculated online by the model, $f(SWC)$ is fitted based on the data curves in Figures
1 and 3 in (Oswald et al., 2013) and the equation is as follows:
$$f(SWC) = 1.04 \times exp^{(-e^{-\frac{SWC-11.32586}{5.27335}} - \frac{SWC-11.32586}{5.27335} + 1)}$$
**2.2 Model experiment design**



The descriptions of the sensitivity simulations are shown in Table S5. Default
simulation uses MEGAN scheme to estimate $SNO_x$ and no SHONO is considered. Base
simulation uses soil Nr emissions schemes with the improvement of using BDISNP
scheme for $SNO_x$ and consideration of SHONO and other four HONO sources (as
described above). Comparison of results from Default and Base simulations is used to
show the improvement in the model performance after updating the soil Nr emissions
schemes and incorporating HONO potential sources. To explore the impact of soil Nr
emissions on $O_3$ and secondary pollutants, we conduct a series of sensitivity simulations
with soil $NO_x$ and HONO emissions turned on/off separately and jointly (anthropogenic
emissions for the year 2017), i.e., NoSoilNr, NoSHONO and $NoSNO_x$. To evaluate the
role of soil Nr emissions on $O_3$ mitigation strategies and air temperature change under
different anthropogenic emission reduction scenarios, we further conduct the
$Base\_redANO_x$ and $NoSoil\_redANO_x$ simulations with anthropogenic $NO_x$ emissions
reduced by 20%, 40%, 60%, 80% and 100%, respectively.
**2.3 Observational data**
The tropospheric column densities of $NO_2$ from TROPOMI (TROPOspheric
Monitoring Instrument) level-2 in version 1 with the horizontal spatial resolution of 3.5
$\times$ 7 $km^2$ are used. The quality controls, i.e., cloud-screened (cloud fraction below 30%)
and quality-assured (qa_value above 0.50), and averaging kernels (AK) are applied in
the comparison of the TROPOMI and UI-WRF-Chem simulated tropospheric $NO_2$
vertical column densities (defined as $NO_2$ VCD).
To evaluate the model performance on simulating surface air pollutants, we use





the hourly surface $O_3$ concentrations at 888 monitoring sites from the China National
Environmental Monitoring Center (CNEMC), and hourly surface HONO
concentrations measured by the In-situ Gas and Aerosol Compositions monitor (IGAC)
(Zhan et al., 2021) at Nanjing University of Information Science & Technology (NUIST)
(32.2º N, 118.7º E; 22m above sea level) (Xu et al., 2019).

**3. Results and discussions**

**3.1 Soil nitrogen emissions and air pollution evaluation**

Figure 1 shows the spatial distribution of simulated monthly mean $SNO_x$ and
SHONO fluxes in July 2018 across North China. In most regions, $SNO_x$ flux is nearly
doubled that of SHONO, and higher $SNO_x$ and SHONO are concentrated in areas
dominated by cropland. The monthly total soil emissions over the whole study domain
(cropland) are 104.5 (82.4) Gg N mon$^{-1}$ for $NO_x$ and 52.7 (45.9) Gg N mon$^{-1}$ for HONO.
In the densely populated BTH region, the monthly total $SNO_x$ are 18.7 Gg N mon$^{-1}$,
which is equivalent to 37.3% of anthropogenic $NO_x$ emissions for the year 2017. For
the FEW region, where also experiences severe $O_3$ and $PM_{2.5}$ pollution, the monthly
total $SNO_x$ (7.0 Gg N mon$^{-1}$) account for 29.2% of anthropogenic $NO_x$ emissions. The
monthly total SHONO in both study regions are much lower than their $SNO_x$
counterparts, with the emissions of 6.9 and 4.6 Gg N mon$^{-1}$, accounting for 13.5% and
19.2% of anthropogenic $NO_x$ emissions in BTH and FWP regions, respectively.
To evaluate the model performance, Figure 2 shows the tropospheric $NO_2$ VCD
from TROPOMI satellite products and UI-WRF-Chem simulations (Default and Base)
during July 2018 in North China. Default and Base can both reproduce the hot spots of



NO$_2$ VCD in urban areas shown in the TROPOMI observations. However, the Default
significantly underestimates the NO$_2$ VCD, especially in regions surrounding urban
areas. It is found that Default underestimates NO$_2$ VCD by 48% over the regions where
soil emissions dominate (i.e., soil Nr emissions contribute more than half to the
atmospheric nitrogen emissions), while the Base reduced the bias to 13% (Figure S2).
Overall, Base shows the improved performance in simulating NO$_2$ VCD in comparison
to Default with a decreasing bias from -30% (-21%) to +4% (+17%) in the study region
(cropland). The overestimated NO$_2$ VCD in Base is most likely attributed to the time
lag in anthropogenic emissions inventory used in the study (Chen et al., 2021),
uncertainties in the stratospheric portion of NO$_2$ VCD and AK caused the retrieval
errors (Van Geffen et al., 2020). Additionally, the estimated SNO$_x$ are also subjected to
certain limitations and uncertainties. The first uncertainty comes from the amount of N
fertilizer application, which has been identified as the dominant contributor to SNO$_x$.
In this study, we use the amount of agricultural N fertilizer application at the province
level from the statistical yearbook to update the default N fertilizer application data in
the model (the baseline year for 2006), but a recent study showed that compound
fertilizer, usually with nitrogen (N), phosphorus (P), and potassium (K), were more
commonly used in China; if only N fertilizer is considered to nudge the N fertilizer
application data in the model, the estimated SNO$_x$ may be underestimated by 11.1%–
41.5% (Huang et al., 2023). Furthermore, although we use the modeled green
vegetation fraction (GVF) to determine the distribution of arid (GVF ≤ 30%) and non-
arid (GVF > 30%) regions. Huber et al. (2023) showed that the estimated SNO$_x$ based



on the static classification of arid vs. non-arid is very sensitive to the soil moisture, and
thus could not produce self-consistent results when using different input soil moisture
products unless a normalized soil moisture index to represent. Therefore, more direct
measurements of soil Nr fluxes are crucial to better constrain soil emissions and
improve the parametrization in the model.

We evaluate the simulation with the surface $O_3$ observations from the China

National        Environmental        Monitoring        Centre        (CNEMC)        network
(http://www.cnemc.cn/en/) (Figure 3). Over the whole study region, the Base can better
capture the spatial distribution of observed surface MDA8 $O_3$ with a relatively higher
spatial correlation of $R = 0.68$ than that in Default ($R = 0.46$). The simulated monthly
averaged MDA8 $O_3$ concentrations across the 888 sites in the study region are 123.0 μg
$m^{-3}$ in Default and 132.5 μg $m^{-3}$ in Base, respectively, which are both slightly higher
than the observed concentrations (120.7 μg $m^{-3}$). Overprediction is also observed for
the FWP and BTH regions in the Base simulation, with the normalized mean bias (NMB)
of 6.1% and 4.9%, respectively (Figure S3). These positive biases are mainly due to
overestimated transport of boundary $O_3$ in both horizontal and vertical directions
(Huang et al., 2023) and underestimated precipitation and cloud cover in the current
model (Sun et al., 2019).

We also compare the simulated surface HONO and nitrate concentrations to the

observations at a rural station in Nanjing during July 2018. Figure 4 shows that the
simulated HONO concentrations in Default are 98.3% lower than the observations. In
comparison, the Base with considering SHONO and other HONO potential sources



significantly improves the simulation performance and reduces the bias to 47.8%, and
also reproduces the diurnal variation of HONO with the temporal correlation of $R =$
0.76. It is worth noting that the concentrations of HONO from 08 am to 18 pm are lower
than the observations, this discrepancy may be attributed to the underestimated
contribution from the predominant sources of HONO during the daytime, such as $NO_2$
heterogeneous reactions on ground and aerosol surfaces. Moreover, the contributions
of different sources to ambient HONO concentrations at this rural station are also
evaluated, the soil emissions could contribute almost 25.8% to the surface HONO,
which may be partially attributed to the high emissions of HONO from croplands
around the city of Nanjing (Figure S4). The results that soil emissions contribute less
to the daytime positive flux than the other source is consistent with previous studies
(Skiba et al., 2020; Wang et al., 2023b). For nitrate concentration, the Base simulation
shows a lower bias (5.6%) and an improved diurnal variation (temporal correlation of
R = 0.92) compared to the Default simulation (bias = 27.8%, R = 0.85). We
acknowledge that there are certain uncertainties in the current model. Nevertheless, the
improved simulation performance compared to the Default illustrates the credibility of
the results obtained from the Base simulation.
**3.2 Impact on $O_3$ formation and air quality**
To quantify the effects of $SNO_x$ and SHONO on atmospheric oxidation capacity,
$O_3$ formation and air quality as well as their combined effect, the conventional brute-
force method was used, i.e., the impact of a specific source is determined in atmospheric
chemistry models as the differences between the standard/base simulation with all




emissions turned on and a sensitivity simulation with this source turned off or perturbed
(Table S5). As shown in Figure 5, the contribution of $SNO_x$ and SHONO to surface
$NO_2$ and HONO has a different spatial pattern from that of the fluxes of $SNO_x$ and
SHONO. Overall, the maximum contribution of $SNO_x$ to the monthly average surface
$NO_2$ concentrations is 78.6%, with a domain-averaged value of 30.3%. Regionally,
$SNO_x$ contribute 5.5 µg m$^{-3}$ (37.1%) and 2.5 µg m$^{-3}$ (31.8%) to the surface $NO_2$ in the
BTH and FWP regions, respectively, which are both higher than the domain-averaged
contribution. Although SHONO fluxes are lower than that of $SNO_x$, its effect on
ambient HONO cannot be ignored. Over the study region, the contribution of SHONO
to surface HONO concentration ranges from 0 to 49.0%, with a domain-averaged value
of 35.6%. For the selected key regions, there are 1.8 µg/m$^3$ (36.7%) and 1.5 µg/m$^3$
(38.0%) of the monthly average HONO concentrations in the BTH and FWP regions,
respectively, from soil emissions. It is noteworthy that, despite the surface $NO_2$ (HONO)
concentrations in the study regions being impacted by less than 13% (17%) due to
SHONO ($SNO_x$), the combined effects of soil Nr emissions on surface $NO_2$ (HONO)
are found to be greater than the individual effects, which are 38.4% (40.3%) for BTH
and 33.9% (40.1%) for FWP region, respectively (Table S6). These results highlight the
importance of considering the cumulative impacts of multiple reactive nitrogen
emissions from soils on air pollution.
Consequently, substantial soil Nr emissions have a non-negligible effect on
atmospheric oxidation and the formation of secondary pollutants. For atmospheric
oxidation, we assess the impact of soil Nr emission on the maximum 1 h (max-1h) ·OH



levels and find that SHONO have a potential to increase the max-1h ·OH in most areas,
with a domain-averaged increase of 10.0%. On the contrary, the inclusion of $SNO_x$
results in a significant reduction of 31.3% in the max-1h ·OH across the entire study
domain. Considering the combined effect of $SNO_x$ and SHONO, there is an overall
decrease of 24.3% in the max-1h ·OH over the study domain, with the BTH region
experiencing a decrease of 22.6% and FWP region showing a relatively greater
reduction of 32.2% (Table S7). These findings are different from the previous study,
which showed that soil background emissions including $NO_x$ and HONO led to a 7.5%
increase in max-1h ·OH in China (Wang et al., 2023b). We stress the crucial role of
$SNO_x$ in influencing ·OH concentrations and highlight the varying impacts across
different regions. For secondary pollutants, substantial $O_3$ enhancement is found in
Henan and Hubei provinces, while the increase in nitrate is consistent with the spatial
pattern of surface $NO_2$ from soil emissions. Specifically, soil Nr emissions increase the
monthly average MDA8 $O_3$ and nitrate concentrations by 18.2% and 31.8%,
respectively, across the study domain, with the increase of 16.9% and 42.4% in the BTH
region and 17.2% and 42.7% in the FWP region. Moreover, soil emissions of $NO_x$ have
a stronger effect on $O_3$ and nitrate in North China than those of SHONO.

The ratio of surface $H_2O_2$ to $HNO_3$ concentrations (hereafter $H_2O_2/HNO_3$) was

used as an indicator of the $O_3$ formation regime to study the changes in sensitivity of
summer $O_3$ to its precursors after considering the soil Nr emissions. The threshold of
$H_2O_2/HNO_3$ for determining $O_3$ formation regime varies regionally (Sillman, 1995),
thus in this study, we identify the regions with $H_2O_2/HNO_3$ values greater than 0.65 as





$NO_x$-sensitive regime, $H_2O_2/HNO_3$ values lower than 0.35 as VOCs-sensitive regime,
and $H_2O_2/HNO_3$ values between 0.35 and 0.65 as VOCs-$NO_x$ mixed sensitive regime
(Shen et al., 2023). Figure 6 illustrates that the majority of BTH region has $H_2O_2/HNO_3$
values lower than 0.35 in Base simulation, indicating a VOCs-sensitive regime or $NO_x$-
saturated regime, which is consistent with the previous studies based on satellite
observations and model simulations (Wang et al., 2019; Wang et al., 2017). The
distribution of sensitivity of $O_3$ to precursor emission in FWP regions are more complex
with a mix of three $O_3$ formation regimes, which is attributed to the large population,
regional urbanization and industrialization. However, when soil nitrogen emissions are
excluded, the $H_2O_2/HNO_3$ values mostly increase within 40% and the $O_3$ formation
regime shifts to VOCs-$NO_x$ mixed sensitive regime and $NO_x$-sensitive regime in both
BTH and FWP regions. Although soil Nr emissions are lower than anthropogenic
emissions, they still could affect the sensitivity of $O_3$ to its precursors and thus have an
impact on the effectiveness of emission reduction policies. Therefore, soil emissions
must be considered in formatting policies for the prevention and management of $O_3$
pollution.
**3.3 Implication on $O_3$ mitigation strategies and temperature rise**

Due to the influence of soil Nr emissions, the sensitivity of $O_3$ pollution to its

precursors varies spatially, depending on the local levels of anthropogenic emissions. It
is thus important to quantify the role of soil Nr emissions in $O_3$ pollution regulation for
improving the effectiveness of air control measures. We conduct a series of sensitivity
experiments with anthropogenic $NO_x$ emissions reduced by 20%, 40%, 60%, 80% and



100%, respectively, relative to the Base simulation (Table S5), and analyze the
difference in the response of surface $O_3$ concentrations to the anthropogenic $NO_x$
emissions reductions in the presence and absence of soil Nr emissions. Figure 7 shows
that with the reduction of anthropogenic $NO_x$ emissions, MDA8 $O_3$ concentrations show
an accelerated decreasing trend, suggesting increasing efficiency of anthropogenic $NO_x$
control measures. And MDA8 $O_3$ response to anthropogenic $NO_x$ emissions in the BTH
region is more curved (nonlinear) than that in the FWP region, which is consistent with
the fact that the BTH tends to have more $NO_x$-saturated $O_3$ production (Figure 6).
It is noted that the reduction of anthropogenic $NO_x$ emissions in the presence of
soil Nr emissions leads to a slower decrease in MDA8 $O_3$ compared to when soil Nr
emissions are excluded. We further analyze the details of the domain-averaged MDA8
$O_3$ changes under different anthropogenic reduction scenarios for the two key regions.
Specifically, in the BTH region, MDA8 $O_3$ decrease by 1.3% (1.8 μg m$^{-3}$), 3.4% (4.6
μg m$^{-3}$), 6.3% (8.7 μg m$^{-3}$), 10.7% (14.7 μg m$^{-3}$), and 17.4% (24.0 μg m$^{-3}$) with
anthropogenic $NO_x$ emission reductions by 20%, 40%, 60%, 80%, and 100%,
respectively, in the present of soil Nr emissions. Comparatively, in the absence of soil
Nr emissions, the reductions in MDA8 $O_3$ are more pronounced and decrease by 2.3%
(2.7 μg m$^{-3}$), 5.6% (6.6 μg m$^{-3}$), 10.7% (12.8 μg m$^{-3}$), 19.4% (23.2 μg m$^{-3}$), and 42.3%
(50.6 μg m$^{-3}$), respectively. In the FWP region, with a 20% reduction in anthropogenic
$NO_x$ emissions, MDA8 $O_3$ levels only exhibit a slight decrease of 1.7% (2.3 μg m$^{-3}$) in
the presence of soil Nr emissions, whereas a decrease of 2.3% (2.6 μg m$^{-3}$) is found in
the absence of soil Nr emissions. When anthropogenic $NO_x$ emissions are removed



entirely, MDA8 $O_3$ decreases by 13.6% (17.7 µg m$^{-3}$) in the presence of soil Nr
emissions, and more significant decreases are found in the absent of soil Nr emissions
with a reduction of 27.4% (34.0 µg m$^{-3}$) (as shown in Figure 7b-c, e-f). We conclude
that the existence of soil Nr emissions could contribute to an additional part of $O_3$
production, amounting to a range of 0-24.9% in the BTH and 0-13.8% in the FWP
region, and these suppressions could be enlarged over the rural areas where have more
substantial soil Nr emissions (e.g. 0-32.3% in cropland over the BTH and 0-15.0% in
croplands over the FWP region). These findings suggest that soil Nr emissions have the
potential to suppress the effectiveness of measures implemented to mitigate $O_3$
pollution, and this effect becomes more significant as anthropogenic emissions increase.
We also quantify the $O_3$ generated from soil Nr emission source (denoted as the
soil $O_3$) under the different anthropogenic $NO_x$ emission scenarios. Overall, soil $O_3$
concentrations in croplands are higher than in non-croplands. Regionally, in the BTH
region, the soil $O_3$ concentrations are 19.8 µg m$^{-3}$ under high anthropogenic emissions
level (referred to as the Base simulation), while the soil $O_3$ concentrations significantly
increase to 46.4 µg m$^{-3}$ when all anthropogenic $NO_x$ emissions are cut down (shown as
red bar in Figure 7b). A similar trend is also found in the FWP region, although soil $O_3$
concentrations are relatively lower than that in the BTH region, the soil $O_3$
concentrations are 19.0 µg m$^{-3}$ in the Base simulation, and do not change significantly
with the reduction of anthropogenic emissions, but increase to 31.9 µg m$^{-3}$ when
anthropogenic $NO_x$ emissions are excluded (shown as red bar in Figure 7c). The
reduction in anthropogenic $NO_x$ emissions results in a shift of the $O_3$ formation regime



towards a more $NO_x$-sensitive chemical regime, leading to a higher contribution of $O_3$
from soil sources. We conclude that with stricter anthropogenic emission reduction
measures, the contributions of soil Nr emissions to $O_3$ production in both absolute and
relative value would increase and further hamper the effectiveness of anthropogenic
emission reductions. To effectively mitigate the desired level of $O_3$ concentrations, it is
necessary to implement much stricter control measures for anthropogenic emissions
due to the synergistic effects of $SNO_x$ and SHONO.

Here we show that the substantial soil Nr emissions present an additional challenge

for $O_3$ pollution regulation in the North China. We further assess the impact of soil Nr
emissions on air temperature change under different anthropogenic emission reduction
scenarios. By comparing changes in air temperature at 2m (T2) with and without soil
Nr emissions under different anthropogenic emission reduction scenarios, Figure 8
shows that incorporating soil Nr emissions results in a slower rate of T2 increase
compared to scenarios without soil Nr emissions, and this phenomenon is consistent
across all study regions. In the FWP region, when anthropogenic $NO_x$ emissions are
eliminated, T2 increases by 0.056 °C in the presence of soil Nr emissions, compared to
0.092 °C in the absence of soil Nr emissions. In the BTH region, which has relatively
high anthropogenic emissions, reducing anthropogenic $NO_x$ emissions by the same
proportion could result in relatively greater warming, and T2 increases by 0.084 °C in
the presence of soil Nr emissions, compared to 0.15 °C in the absence of soil Nr
emissions when anthropogenic $NO_x$ emissions are excluded. This is attributed to
aerosols (such as sulfate and nitrate) and $NO_x$ emissions and their effective radiative



forcing (ERF) associated with a cooling effect (high confidence) (Liao and Xie, 2021;
Bellouin et al., 2020). Decreases in aerosol concentrations and $NO_x$ emissions could
weaken the cooling effect and potentially accelerate warming to some extent, while soil
Nr emissions can offset temperature rise caused by declining anthropogenic $NO_x$
emissions (Figure S5). Therefore, although soil Nr emissions are relatively low
compared to anthropogenic emissions, the combined effects of $NO_x$ and HONO
emissions from natural soil and agricultural land should be considered when assessing
climate change and implementing strategies to mitigate $O_3$ pollution.
**4. Conclusions**

In this study, the updated soil Nr emission scheme was implemented in the UI-

WRF-Chem model and used to estimate the combined and individual impact of $SNO_x$
and SHONO on subsequent changes in air quality and air temperature rise in North
China, with a focus on two key regions (the BTH and FWP regions) because of high
levels of soil Nr and anthropogenic emissions. We show that the $SNO_x$ flux is nearly
doubled that of SHONO during July 2018, with higher soil emissions in areas with
extensive cropland. The contribution of soil Nr emissions to average surface $NO_2$ and
HONO are 38.4% and 40.3% in the BTH, and 33.9% and 40.1% in the FWP region,
respectively, and the substantial soil Nr emissions lead to a considerable increase in the
monthly average MDA8 $O_3$ and nitrate concentrations, with the values of 16.9% and
42.4% in the BTH region and 17.2% and 42.7% in the FWP region, which both exceed
the individual $SNO_x$ or SHONO effect. The presence of soil Nr emissions, acting as
precursors of $O_3$ and secondary inorganic aerosols, has a suppressing effect on efforts



to mitigate $O_3$ pollution, particularly in the BTH region, and also leads to a slower
increase rate of T2 compared to scenarios without soil Nr emissions. We note that the
effect of soil Nr emissions shows spatial heterogeneity under different anthropogenic
$NO_x$ emissions reduction scenarios.

However, we admit that uncertainties in both soil Nr and anthropogenic emissions,

as well as the parameterization scheme of HONO sources. The agricultural emissions
of another important reactive nitrogen gas, $NH_3$, may also be underestimated due to
uncertainties in agricultural fertilizer application and livestock waste in MEIC
inventory (Li et al., 2021a). These uncertainties could impact the aerosol formation and
local cooling effect. Also, the discrepancies between simulated and observed $NO_2$, $O_3$
and other air pollutants in the model may affect the assessment of the role of soil Nr
emissions in $O_3$ mitigation strategies and their impact on climate change. Thus, more
direct measurements of soil Nr fluxes are crucial to better constrain soil emissions and
improve the parametrization in the model.

Our study highlights that despite soil Nr emissions being lower than anthropogenic

emissions, they still have a substantial impact on the effectiveness of $O_3$ pollution
mitigation measures, and this effect becomes more significant as anthropogenic
emissions decrease. Therefore, reactive nitrogen from soil emission source must be
considered in formatting measures for the prevention and management of $O_3$, as well
as addressing climate change.



*Code and data availability.* Some of the data repositories have been listed in Section 2.
The other data, model outputs and codes can be accessed by contacting Tong Sha via
tong-sha@sust.edu.cn.
*Author contributions.* TS performed the model simulation, data analysis and
manuscript writing. TS and JW proposed the idea. SY, QC and LL supervised this work
and revised the manuscript. XM, ZF and KB helped the revision of the manuscript. YZ
provided and analyzed the observation data.
*Competing interests.* The authors declare that they have no conflict of interest.
*Acknowledgements.* This study is supported by the National Natural Science
Foundation of China (grant nos. 42205107, 42130714). Jun Wang's participation is
made possible via the in-kind support from the University of Iowa.

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



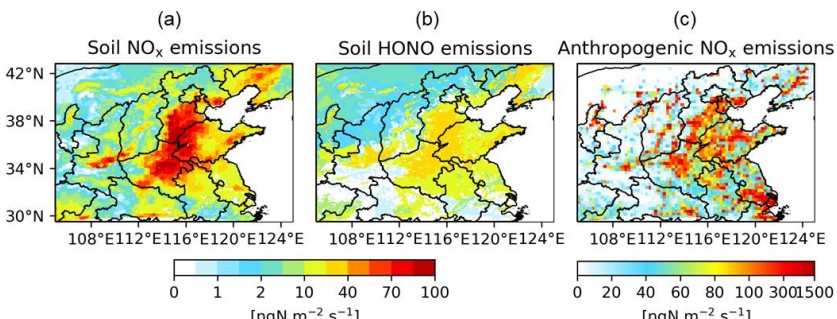

**Figure 1.** Distribution of the simulated monthly mean (a) soil NO$_x$ emissions, (b) soil

HONO emissions, and (c) anthropogenic NO$_x$ emissions in July 2018.



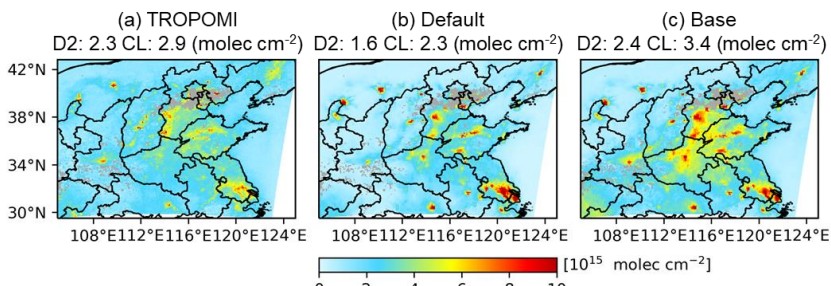

**Figure 2.** (a) Monthly mean tropospheric $NO_2$ VCD retrieved by TROPOMI measured at 12:00−14:00 LT and simulated by (e) Default and (f) Base averaged over the same periods in July 2018.



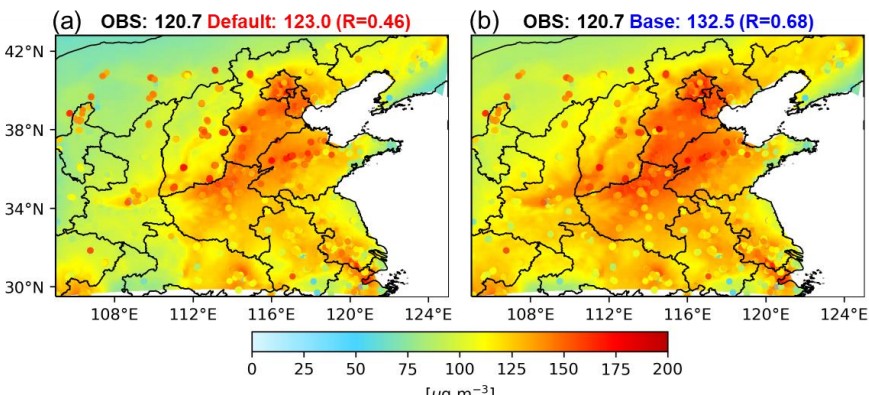

**Figure 3.** Distribution of observed (dots) and simulated (shaded) surface MDA8 $O_3$

from (a) Default and (b) Base in July 2018. Statistics in the upper corner of panels are

the monthly mean MDA8 $O_3$ concentrations averaged over the study region and the

spatial correlation coefficient R between observations and simulations.





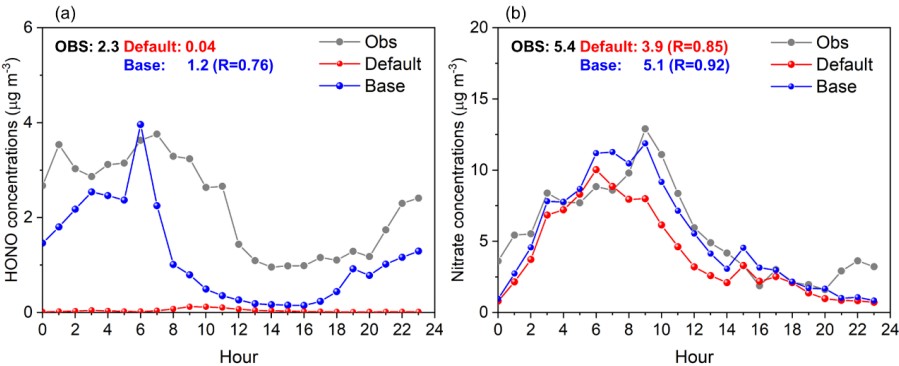

768

**Figure 4.** Diurnal variation of observed (in grey) and simulated (Default in red and

Base in blue) surface (a) HONO and (b) nitrate concentrations at a rural station in

Nanjing, with the mean value and temporal correlation coefficients (R) shown in the

upper right corner.

773

774



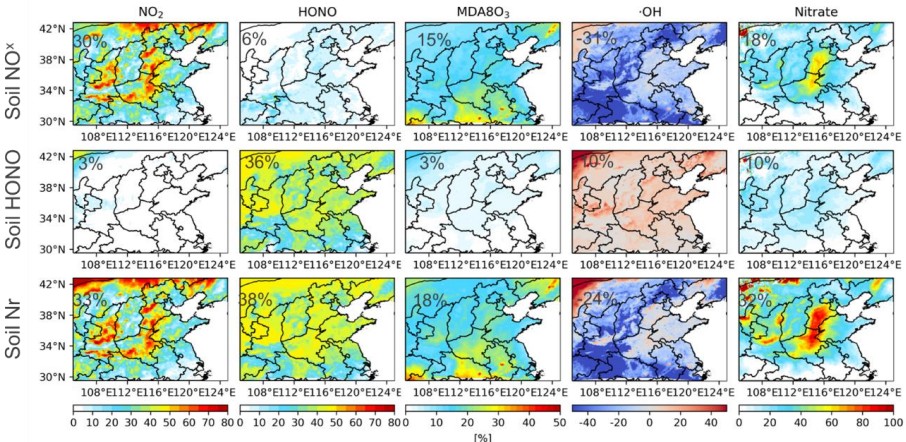

**Figure 5.** Simulated effects of soil Nr emissions on air quality. The first and second rows show the contributions of soil NO$_x$ and soil HONO emissions on monthly average concentrations of NO$_2$, HONO, MDA8O$_3$, max-1h ·OH, and nitrate, respectively. The third row shows the combined effect of soil Nr emissions on the species listed above. Statistics in the upper right corner of each panel are the mean values averaged over the study region.



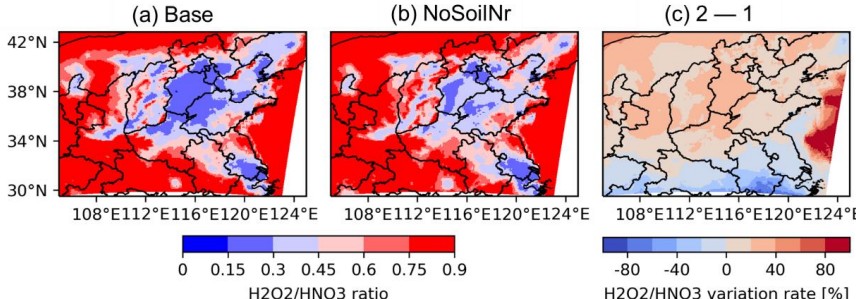

**Figure 6.** Distribution of the $O_3$ formation regimes (represented as $H_2O_2/HNO_3$ ratios)

for (a) Base simulation with the addition of soil Nr emissions and (b) NoSoilNr

simulation without the addition of soil Nr emissions. (c) Changes in the distribution of

$O_3$ formation regimes due to the soil Nr emissions





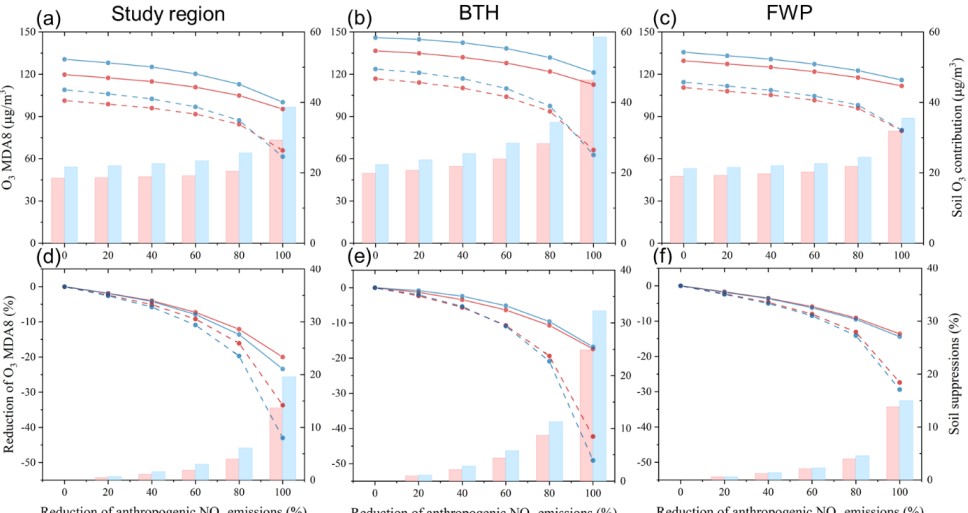

789

**Figure 7.** Role of soil Nr emissions in O₃ pollution regulation. The responses of MDA8

O₃ concentrations to the reductions of anthropogenic NOₓ emissions (20%, 40%, 60%,

80% and 100%) relative to July 2018 levels, in the presence (solid line) and absence

(dotted line) of soil Nr emissions in the study region, BTH and FWP region. (The lines

in panels a-c and d-f are MDA8 O₃ concentrations and the relative reductions in MDA8

O₃ under different anthropogenic NOₓ emission reductions, respectively. The red bars

(right y-axis) in panels a-c show the corresponding O₃ contribution from soil Nr

emissions, which is determined as the difference between the solid and dotted lines, and

the blue bars are the same as the red bars but for statistics in cropland. The red bars

(right y-axis) in panels d-f show the suppression of O₃ pollution mitigated due to the

existence of soil Nr emissions, which are determined as the difference between the solid

and dotted lines, and the blue bars are the same as the red bars but for statistics in

cropland.)

803



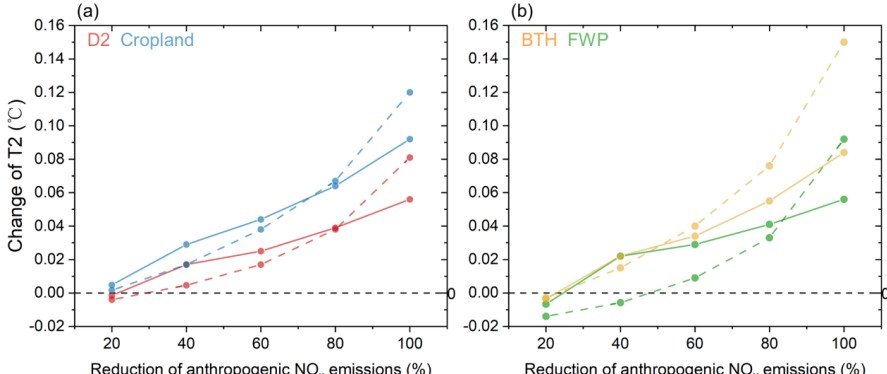

**Figure 8.** The responses of air temperature at 2m (T2) to the reductions of anthropogenic NO$_x$ emissions (20%, 40%, 60%, 80% and 100%) relative to July 2018 levels in the presence (solid line) and absence (dotted line) of soil Nr emissions (a) in the study region, (b) BTH and FWP region.