# Peer review of "Large contribution of soil emissions to the atmospheric nitrogen"

_EGUsphere, 2024_

## Author Comment (AC1)

**Reply to Reviewer#1 and the editor:**

We thank the reviewers for taking the time to assess the manuscript (*egusphere-2024-359*) and providing helpful comments and suggestions to improve the manuscript. Below we address the reviewers' comments, with the reviewer comments in *italic and black*, and our responses in blue. We have revised the manuscript accordingly and mentioned the line number in the **tracked version of the manuscript**.

*This manuscript focuses on the effects of soil nitrogen emission (SNOX and SHONO) on ozone pollution over two typical regions (BTH and FWP) in China. The authors found that SNOx and SHONO emissions account for a large proportion (nearly 50%) of total anthropogenic NOx emissions. Therefore, it can increase MDA8 O3 by 17 % and nitrate concentrations by 42% in the BTH. The results suggest that the presence of soil nitrogen emissions can offset the efforts in controlling anthropogenic emissions, which can provide theoretical implications on ozone pollution management in key regions of China. Overall, the manuscript is well written, although minor changes are needed to further improve it.*

Thanks for the positive comments. Our item-by-item responses can be found in below.

*Comments:*

*It is not clear how MEIC emissions are converted to model-ready formats.*

Thank you for the suggestion regarding the conversion of MEIC emissions to model-ready formats. The MEIC inventory used in this study has a spatial resolution of $0.25\,°\times0.25\,°$and is constructed on an equal latitude-longitude grid. The model domain, however, employs a Lambert projection, which results in a misalignment between these two grid systems. To address this issue, we implement a spatial interpolation method to reallocate emission fluxes to the model grid. Here is a detailed description of our method:

Under the Lambert projection, the model grids are rectangular, while the MEIC grids are deformed, approximating a trapezoid shape. For calculating the emission of each model grid, we determine which MEIC grid may fall in that grid, based on their central latitude and longitude coordinates, and then apply the principle that the ratio of emissions is equivalent to the ratio of areas between the model grid and MEIC grid. The area of the MEIC grid is denoted as A and its corresponding emission is denoted as E, and the area of the model grid is denoted as a, thus the emission fluxes of model grid e are expressed as $e = E \times a / A$. However, if the spatial resolution of the simulation domain is coarser than MEIC, the model grid often falls within multiple MEIC grids and this method would have errors. We thus divide the coarser model grid into $n \times n$ finer subgrids. Since the spatial resolution of the nested domain is 9 km in this study, we choose n as 9. We apply the aforementioned method to calculate the emissions of

each finer model subgrid, and then sum up the emissions of n × n finer subgrids to obtain the total emissions of the model grid. This method ensures a more accurate allocation of MEIC emission to the model grid, despite different spatial resolutions of the simulation domain.

The description of the above conversion method has also been added in the revised version as follows:

Page 8, Lines 144-146: "Due to the differences in spatial resolution and map projection between the MEIC inventory and model grid, we applied a spatial interpolation method to convert the MEIC inventory to the model-ready formats. The descriptions are detailed in Text S1."

Please see the detailed description in Text S1 in the revised Supplement.

*Is there any way to validate the emissions of NOx and HONO using BDISNP?*

We admit that the optimal validation method is to compare the simulated and observed soil emission flux of $NO_x$ and HONO. Due to the influence of various factors, the observed soil nitrogen emissions reported in the literature couldn't align well with the simulated $SNO_x$ and SHONO. For example, many field observation experiments are conducted to study the impacts of anthropogenic fertilizer application and irrigation on soil Nr emissions, thus the measured soil Nr emission data are not representative in comparison to the model results (Li, 2013; Tang et al., 2020; Xue et al., 2019). Additionally, the observed $SNO_x$ and SHONO in the literature are often specific to certain regions and are based on a relatively limited number of observation sites (Wu et al., 2019; Wu et al., 2022). Therefore, it is difficult to compare the scattered and location-specific observations of soil Nr emissions reported in the literature (from different regions and periods) with the UI-WRF-Chem model results. To evaluate the simulation performance of the BDISNP scheme, we use satellite observations that have large spatial coverage and compare them with the simulated concentrations of $NO_2$, HONO, $O_3$, and nitrate in the atmosphere with and without the implementation of the soil Nr emission scheme against the observations. From Figure 2 to 4, the results show that the simulations with the implementation of the BDISNP scheme are in better agreement with TROPOMI $NO_2$ VCD, surface HONO, $O_3$, and nitrate than the Default. Similar approach of validation was used in our past work as well (Sha et al., 2021).

*In Figure 2, please show statistical values of the comparison. It is hard to say which is better in comparison to the TROPOMI results.*

We have added the normalized mean bias (NMB) and spatial correlation coefficient (R) in Figure 2, and also modified the discussions in the revised version as follows:

Page 14, Lines 283-286: "Overall, Base shows the improved performance in

simulating NO$_2$ VCD in comparison to Default with a decreasing bias from -30% (-21%) to +4% (+17%) and an increasing spatial correlation coefficient (R) from 0.62 (0.50) to 0.65 (0.54) in the study region (cropland).”

*Figure 6, the panel (c) is not correctly named.*

Thanks for the reminder, the title of panel (c) in Figure 6 has been corrected in the revised manuscript.

*In Conclusion, "leads to a slower increase rate of T2 compared to scenarios without soil Nr emissions", I suggest add the values of T2 increase rate in parentheses (??℃).*

We have added the values of T2 increase rate in the revised version as follows:

Page 26, Lines 547-551: “The presence of soil Nr emissions, acting as precursors of O$_3$ and SIA, has a suppressing effect on efforts to mitigate summer O$_3$ pollution, particularly in the BTH region, and also leads to a slower increase rate of T2 (0.098 ℃) in July compared to scenarios without soil Nr emissions (0.14 ℃) when anthropogenic emissions are excluded.”

*Line 332-351, the authors found that soil NOX lead to OH decrease in their study, which is contrary to other studies that showed soil NOX can increase OH. I suggest add some explanations.*

Thanks for the suggestion. We have added the discussions in the revised version as follows:

Page 19, Lines 395-403: “The discrepancy between our findings and those of other studies regarding the impact of SNO$_x$ on ·OH levels could be attributed to the abundance of ambient NH$_3$ in China during summer, where soil emissions may lead to a significant increase in nitrate, and the increased aerosols can affect the concentrations of ·OH through photochemical reactions (Wang et al., 2011; Xu et al., 2022). Additionally, after taking into account the SNO$_x$ in the model, the environment may shift to a relatively NO$_x$-saturated regime, thus the termination reaction for O$_3$ production could be NO$_2$ and ·OH to generate HNO$_3$ (Chen et al., 2022; Wang et al., 2023b).”

*Line 363-365, "The distribution of sensitivity of O3 to precursor emission in FWP regions are more complex with a mix of three O3 formation regimes, which is attributed to the large population, regional urbanization and industrialization". The explanations are broad, I suggest the authors give more specific explanations or cite some relevant papers, otherwise no need to explain this phenomenon.*

We have modified the explanations in the revised version as follows:

Page 20-21, Lines 419-430: "Figure 6 illustrates that the majority of BTH region has $H_2O_2/HNO_3$ values lower than 0.35 in Base simulation, indicating a VOCs-sensitive regime or $NO_x$-saturated regime in July. In contrast, the distribution of sensitivity of $O_3$ to precursor emission in FWP regions is more complex with a mix of three $O_3$ formation regimes. The spatial patterns of $O_3$ formation regimes presented in this study are consistent with the previous studies based on satellite observations and model simulations during summer seasons, despite using a different method (Wang et al., 2019; Wang et al., 2023b). This agreement across multiple approaches strengthens the confidence in the spatial patterns of $O_3$ formation regimes in the key regions of China."

**Corrections:**

*The reference "TROPOMI ATBD of the total and tropospheric NO2 data products", please check it.*

We have revised this reference to "van Geffen, J. H. G. M., Eskes, H. J., Boersma, K. F., and Veefkind, J. P.: TROPOMI ATBD of the total and tropospheric $NO_2$ data products, Report S5P-KNMI-L2-0005-RP, version 2.2.0, 2021-06-16, KNMI, De Bilt, The Netherlands, http://www.tropomi.eu/data-products/nitrogen-dioxide/ (last access: 7 March 2022), 2021." (Page 34, Lines 761-765)

*In Line 471, "we admit that uncertainties in both soil Nr and anthropogenic emissions", add "exist" after "uncertainties"*

Thanks, we have corrected it in the revised version. (Page 26, Lines 553)

*In Figure 2 caption, note that (e) and (f) should be (b) and (c)*

Thanks, we have corrected it. See Figure 2 in the revised version.

*Line 228, "(Zhan et al., 2021)", maybe Zhang et al?*

We forgot to cite this literature, and have added "Zhan, Y., Xie, M., Gao, D., Wang, T., Zhang, M., and An, F.: Characterization and source analysis of water-soluble inorganic ionic species in $PM_{2.5}$ during a wintertime particle pollution episode in Nanjing, China, Atmos. Res., 262, 105769, 10.1016/j.atmosres.2021.105769, 2021." in the reference list. (Page 37, Lines 855-858)

*Line 391-394, no need to entirely list values under 20%, 40%, 60%, 80%, and 100%. Maybe three percent (20%,60%,100%) are enough.*

We have revised it to "Specifically, in the BTH region, MDA8 $O_3$ decrease by 1.3% (1.8 μg m$^{-3}$), 6.3% (8.7 μg m$^{-3}$), and 17.4% (24.0 μg m$^{-3}$) with anthropogenic $NO_x$ emission reductions by 20%, 60%, and 100%, respectively, in the present of soil Nr emissions." (Page 22, Lines 457-460)

References:

Chen, W., Guenther, A. B., Jia, S., Mao, J., Yan, F., Wang, X., and Shao, M.: Synergistic effects of biogenic volatile organic compounds and soil nitric oxide emissions on summertime ozone formation in China, Sci. Total Environ., 828, 154218, 10.1016/j.scitotenv.2022.154218, 2022.

Li, D.: Emissions of NO and $NH_3$ from a Typical Vegetable-Land Soil after the Application of Chemical N Fertilizers in the Pearl River Delta, PLoS One, 8, e59360, 10.1371/journal.pone.0059360, 2013.

Sha, T., Ma, X., Zhang, H., Janechek, N., Wang, Y., Wang, Y., Castro García, L., Jenerette, G. D., and Wang, J.: Impacts of Soil $NO_x$ Emission on $O_3$ Air Quality in Rural California, Environ. Sci. Technol., 55, 7113-7122, 10.1021/acs.est.0c06834, 2021.

Tang, K., Qin, M., Fang, W., Duan, J., Meng, F., Ye, K., Zhang, H., Xie, P., Liu, J., Liu, W., Feng, Y., Huang, Y., and Ni, T.: An automated dynamic chamber system for exchange flux measurement of reactive nitrogen oxides (HONO and $NO_x$) in farmland ecosystems of the Huaihe River Basin, China, Sci. Total Environ., 745, 140867, 10.1016/j.scitotenv.2020.140867, 2020.

Wang, N., Lyu, X., Deng, X., Huang, X., Jiang, F., and Ding, A.: Aggravating $O_3$ pollution due to $NO_x$ emission control in eastern China, Sci. Total Environ., 677, 732-744, 10.1016/j.scitotenv.2019.04.388, 2019.

Wang, S., Xing, J., Jang, C., Zhu, Y., Fu, J. S., and Hao, J.: Impact Assessment of Ammonia Emissions on Inorganic Aerosols in East China Using Response Surface Modeling Technique, Environ. Sci. Technol., 45, 9293-9300, 10.1021/es2022347, 2011.

Wang, W., Li, X., Cheng, Y., Parrish, D. D., Ni, R., Tan, Z., Liu, Y., Lu, S., Wu, Y., Chen, S., Lu, K., Hu, M., Zeng, L., Shao, M., Huang, C., Tian, X., Leung, K. M., Chen, L., Fan, M., Zhang, Q., Rohrer, F., Wahner, A., Pöschl, U., Su, H., and Zhang, Y.: Ozone pollution mitigation strategy informed by long-term trends of atmospheric oxidation capacity, Nat. Geosci., 17, 20-25, 10.1038/s41561-023-01334-9, 2023b.

Wu, D., Horn, M. A., Behrendt, T., Müller, S., Li, J., Cole, J. A., Xie, B., Ju, X., Li, G., Ermel, M., Oswald, R., Fröhlich-Nowoisky, J., Hoor, P., Hu, C., Liu, M., Andreae, M. O., Pöschl, U., Cheng, Y., Su, H., Trebs, I., Weber, B., and Sörgel, M.: Soil HONO emissions at high moisture content are driven by microbial nitrate reduction to nitrite: tackling the HONO puzzle, ISME J., 13, 1688-1699, 10.1038/s41396-019-0379-y, 2019.

Wu, D., Zhang, J., Wang, M., An, J., Wang, R., Haider, H., Xu-Ri, Huang, Y., Zhang, Q., Zhou, F., Tian, H., Zhang, X., Deng, L., Pan, Y., Chen, X., Yu, Y., Hu, C., Wang, R., Song, Y., Gao, Z., Wang, Y., Hou, L., and Liu, M.: Global and Regional Patterns of Soil Nitrous Acid Emissions and Their Acceleration of Rural Photochemical Reactions, J. Geophys. Res.: Atmos., 127, 10.1029/2021jd036379, 2022.

Xu, W., Zhao, Y., Wen, Z., Chang, Y., Pan, Y., Sun, Y., Ma, X., Sha, Z., Li, Z., Kang, J., Liu, L., Tang, A., Wang, K., Zhang, Y., Guo, Y., Zhang, L., Sheng, L., Zhang, X., Gu, B., Song, Y., Van Damme, M., Clarisse, L., Coheur, P.-F., Collett, J. L., Goulding, K., Zhang, F., He, K., and Liu, X.: Increasing importance of ammonia emission abatement in $PM_{2.5}$ pollution control, Sci. Bull., 67, 1745-1749,

10.1016/j.scib.2022.07.021, 2022.

Xue, C., Ye, C., Zhang, Y., Ma, Z., Liu, P., Zhang, C., Zhao, X., Liu, J., and Mu, Y.: Development and application of a twin open-top chambers method to measure soil HONO emission in the North China Plain, Sci. Total Environ., 659, 621-631, 10.1016/j.scitotenv.2018.12.245, 2019.

---

## Author Comment (AC2)

**Reply to Reviewer#2 and the editor:**

We thank the reviewers for taking the time to assess the manuscript (*egusphere-2024-359*) and providing helpful comments and suggestions to improve the manuscript. Below we address the reviewers' comments, with the reviewer comments in *italic and black*, and our responses in blue. We have revised the manuscript accordingly and mentioned the line number in the **tracked version of the manuscript**.

*The authors estimate both soil NOx and HONO emissions in North China and investigate their impacts on air quality and temperature using an updated soil Nr emissions scheme within a chemical transport model. The inclusion of this scheme appears to significantly impact the model's outputs. I recommend the paper for publication, subject to the following revisions:*

Thanks for the positive comments. Our item-by-item responses can be found in below.

*General comment:*

*1. Simulation Settings Summary:*

*Please consider adding a table summarizing the settings for each simulation scenario to enhance clarity and ease of comparison.*

Thanks for the suggestion. The table summarizing the settings for each simulation scenario has been moved from the Supplement to the revised manuscript. Additionally, the descriptions of emission reduction scenarios for co-emitted air pollutants, as you suggested, have also been included in Table 1 of the revised version, and the corresponding description has also been added in the revised manuscript as follows:

Page 11-12, Lines 224-236: "To investigate the relative importance and interaction between anthropogenic and natural emissions of nitrogen-containing pollutants, we conduct the Base_redANO$_x$ and NoSoil_redANO$_x$ simulations to evaluate the role of soil Nr emissions on O$_3$ mitigation strategies, in which anthropogenic NO$_x$ emissions reduced by 20%, 40%, 60%, 80%, and 100%, respectively. Furthermore, considering the co-control of multiple air pollutants and greenhouse gas reductions in future emission reduction scenarios, the Base_redAnt and NoSoil_redAnt simulations are conducted to evaluate the role of soil Nr emissions on air temperature change, and the anthropogenic reduction scenarios simultaneously consider SO$_2$, NO$_x$, primary PM$_{2.5}$, VOCs, and CO emissions reductions (reduced by 20%, 40%, 60%, 80%, and 100%)."

*2. Study Duration and Selection of Year:*

*The investigation is limited to July 2018. This limited scope raises concerns about the generalizability of the conclusions regarding the impact of soil Nr emissions. Could the*

*authors clarify the reasons behind choosing only 2018? Additionally, how do the 2018 temperature, precipitation, and other relevant meteorological factors compare with other years? Given the close relationship between soil emissions and meteorological conditions, it would be beneficial to include additional years to demonstrate the sensitivity to varying weather conditions. The impact of soil Nr emissions during other months should also be discussed, as the atmospheric nitrogen budget from soil emissions is expected to be significantly different at other times of the year.*

The July 2018 chosen as the study period is based on several factors. Firstly, it is reasonable to choose summer because $O_3$ pollution is serious during this time and soil nitrogen emissions are generally the highest. Shen et al. (2023) used multi-source remote sensing observations to optimize the input data and estimated soil $NO_x$ emissions based on the modified YL95 scheme (Yienger and Levy, 1995). Their study also confirmed that the peak in $SNO_x$ from 2017 to 2019 occurred in July, despite the emission estimated methods being different from ours. This conclusion further supports the reason for choosing July as the representative month for soil Nr emission research. The exclusion of the years 2020-2022 from the study period is due to the COVID-19 pandemic, which could affect anthropogenic emissions, atmospheric oxidizing capacity, and meteorological conditions (Liu, 2020; Le et al., 2020; Wang et al., 2022a). Given the potential impact of the COVID-19 pandemic on the representativeness of the results, three pandemic years (2020-2022) are not considered as the study period. Furthermore, previous studies showed that the MDA8 $O_3$ concentrations in China continued to increase during the warm seasons from 2014 to 2017, and then at a slower increase rate from 2018 to 2020, nevertheless the MDA8 $O_3$ was still high in the year 2018-2020 (Liu et al., 2023; Wang et al., 2022b; Yin et al., 2021). We also analyzed the MDA8 $O_3$ concentrations from CNEMC data during the warm seasons in the BTH and FWP regions, and found a fluctuating upward trend from 2017 to 2019 with relatively small variation (Fig. R1). From the perspective of $O_3$ pollution, thus selecting the year 2018 as the study period is representative. Additionally, we also analyzed the air temperatures at 2m (T2) and total precipitations from the MERRA-2 dataset from June to August in 2018. It was shown that higher temperatures and more frequent precipitation occurred in July over the study region (Fig. R2 and R3). Based on the above analysis, we chose July 2018 as the study period.

The reason for selecting July 2018 as the study period has been added in the revised version as follows:

Page 6-7, Lines115-119: "July 2018 was chosen as the study period because of severe $O_3$ pollution during this month, as well as higher air temperatures and more frequent precipitation compared to June and August (Figure S1 and S2), which could contribute to enhanced the soil Nr emissions (Figure S3)."

Soil Nr emissions are significantly influenced by meteorological factors, especially the temperature, humidity, and precipitation. We thus compared the monthly average T2 and total precipitation from the MERRA-2 dataset during summer seasons

in 2017-2019 (Fig. R2 and R3). The results showed that there were no significant variations in the meteorological conditions during these three years in the study region. The monthly average T2 in summer seasons in 2017-2019 showed only slight changes, with ranges of 0.7 ℃ for June, 1.2 ℃ for July, and 1.1 ℃ for August, respectively; the total precipitation in each month was also slightly different, with ranges of 19.9 mm in June, 39.0 mm in July, and 19.3 mm in August, respectively.

Despite considering additional years would provide insights into the sensitivity of soil emissions to varying meteorological conditions, it is noted that our study focused on studying the contribution of soil emissions to the atmospheric nitrogen budget and their impacts on air quality and temperature rise in North China, thus the representative year and month were selected to investigate. Additionally, Tan et al. (2023) also highlighted that the increased impact of soil Nr emissions on $O_3$ contribution was not primarily driven by weather-induced increases in soil Nr emissions, but by the concurrent decreases in fuel combustion $NO_x$ emissions, which enhanced $O_3$ production efficiency from soil by pushing $O_3$ production toward a more $NO_x$-sensitive regime.

To accept your suggestion, we have conducted simulations in January, April, and October 2018, representing winter, spring, and autumn, respectively. As shown in Figure R4, the soil Nr emissions in July are much higher than the other seasons due to higher air temperatures and frequent precipitation, accounting for 39.5% of anthropogenic $NO_x$ emissions over the study region, and 50.2% in the BTH, 47.4% in FWP. Given the substantial contribution of soil emissions to the atmospheric nitrogen budget in July, it is reasonable to expect that soil Nr emissions have a more significant impact on air quality during this season. Moreover, previous studies on soil nitrogen emissions have also focused on summer (Huang et al., 2023; Shen et al., 2023; Wang et al., 2023c), thus we choose the most representative month to assess the impact of soil Nr emissions on air quality and climate change.

The relevant discussions have been added to the revised manuscript as follows:

Page 13 Lines 252-262: "The soil Nr emissions in July are much higher than the other seasons due to higher air temperatures and frequent precipitation, accounting for 39.5% of anthropogenic $NO_x$ emissions over the study region, and 50.2% in the BTH, 47.4% in FWP, which is consistent with the previous studies (Huang et al., 2023; Shen et al., 2023; Wang et al., 2023c). And the proportions can increase to 58.9%, 57.0%, and 65.0%, respectively, when only statistics over the cropland in these regions (Figure S3). Given the substantial contribution of soil emissions to the atmospheric nitrogen budget in July, we thus choose this month to assess the impact of soil Nr emissions on air quality and climate change."

[Figure]

Figure R1. Variation trends of the monthly average MDA8 O₃ concentrations during warm seasons (April-September) in 2014-2023 from the China National Environmental Monitoring Center (CNEMC) over the BTH and FWP regions. The statistics over the x-axis and upper left corner are the warm-season averaged values and the absolute annual linear trend of MDA8 O₃ concentrations (±standard deviation).

[Figure]

Figure R2. Distribution of (a) the monthly average air temperatures at 2m (T2) and (b) total precipitation from the MERRA-2 dataset during June-August in 2017-2019. The statistics in the lower left corner are the monthly average T2 and total precipitation in China and the study region.

[Figure]

Figure R3. Frequency of (a) the monthly average air temperatures at 2m (T2) and (b) total precipitation over the study region during June-August in 2017-2019. The statistics on each panel are the values of T2 and total precipitation amount corresponding to the highest frequency.

[Figure]

Figure R4. Monthly proportion of soil Nr emissions to anthropogenic $NO_x$ emissions during January, April, July, and October in the study region, BTH, and FWP regions. The darker columns with borders are statistics for the whole region, while the lighter columns are statistics for croplands. The gray horizontal dotted line in the figure represents a 50% proportion.

*3. Figure 3 Analysis:*

*The base scenario improves correlation but introduces larger biases, particularly when compared with TROPOMI NO2 data. A detailed discussion regarding the causes of these biases would be valuable. I notice that the performance of the base scenario is better than the default one in Figure 4. If it is the most important justification for the better performance of "base", I suggest additional clarification to justify why HCHO validation weights are more important than NO2 and O3 for this study.*

For the simulation evaluation of $NO_2$ VCD in Figure 2, it is noted that the retrieval of TROPOMI $NO_2$ also has certain uncertainty. The instantaneous uncertainty of TROPOMI tropospheric $NO_2$ columns at the pixel level is 25-50% or can be up to $0.5\sim0.6\times10^{15}$ molecules $cm^{-2}$ (Van Geffen et al., 2020; Van Geffen et al., 2021). The results of Base simulation are in better agreement with TROPOMI than the Default with a reduced NMB and an increased spatial correlation coefficient (R). Additionally, the uncertainty can be random, and our focus here is to reduce the bias between the simulations and observations. We have added the statistical values in Figure 2, and also added corresponding discussions in the revised version as follows:

Page 14-15, Line 283-295: "Overall, Base shows the improved performance in simulating $NO_2$ VCD in comparison to Default with a decreasing bias from -30% (-21%) to +4% (+17%) and an increasing spatial correlation coefficient (R) from 0.62 (0.50) to 0.65 (0.54) in the study region (cropland). However, there is still a discrepancy between the Base simulation and TROPOMI $NO_2$ VCD. This discrepancy could be driven by the combined effects from uncertainties in simulations and observations, associated with the time lag in anthropogenic emissions inventory used in the model (Chen et al., 2021), instantaneous uncertainties in TROPOMI tropospheric $NO_2$ VCD at the pixel level (up to 25-50% or $0.5\sim0.6\times10^{15}$ molecules $cm^{-2}$), as well as uncertainties of stratospheric portion of $NO_2$ VCD and AK caused the retrieval errors (Van Geffen et al., 2020; Van Geffen et al., 2021)."

Page 15, Line 311-314: "Nevertheless, the improved simulation performance of $NO_2$ VCD with a reduced bias and increased spatial correlation coefficient in Base is credible, and soil Nr emission scheme has the fidelity needed to study the implication of soil Nr emissions to air quality in North China."

For the simulation evaluation of MDA8 $O_3$ in Figure 3, we admit there are inherent discrepancies between MDA8 $O_3$ simulation and observation in the Default. Sources of $O_3$ biases in chemical transportation models (CTMs) are complex and multifaceted, which may arise from simplifications of complex chemical mechanisms and physical processes such as dry deposition and vertical mixing (Akimoto et al., 2019; Travis and Jacob, 2019). Input data, including emission inventories, meteorological fields, and other parameters, also tend to be biased (Sun et al., 2019; Ye et al., 2022). Previous studies showed that the mean normalized biases (NMB) of simulated $O_3$ concentrations were within ±30% for nearly 80% of the cases collected from air quality model studies.

The atmospheric chemical transport models like GEOS-Chem and CAMx commonly overestimated the ambient $O_3$ concentration, while the biases for CMAQ, WRF-Chem, and NAQPMS were less conclusive (Yang and Zhao, 2023). These suggest a potential systematic $O_3$ bias in the CTMs. In Base simulation, the inclusion of soil Nr emissions can promote the $O_3$ formation, resulting in higher $O_3$ concentrations and larger deviations. Nevertheless, the increased spatial correlation and reasonable bias found in the Base indicate that the application of the soil Nr emission schemes can effectively improve the simulation performance of MDA8 $O_3$.

Detailed discussions regarding the causes of $O_3$ biases have been added in the revised version as follows:

Page 16-17 Lines 327-337: "Previous studies showed that the NMB of simulated $O_3$ concentrations were within ±30% for nearly 80% of the cases collected from air quality model studies (Yang and Zhao, 2023). These discrepancies may arise from simplifications of complex chemical mechanisms and physical processes, such as dry deposition and vertical mixing (Akimoto et al., 2019; Travis and Jacob, 2019). The uncertainties of input data, including emission inventories, meteorological fields, and other parameters, may also contribute to these discrepancies (Sun et al., 2019; Ye et al., 2022), suggesting a potential systematic $O_3$ bias in air quality models. Therefore, the increased spatial correlation and reasonable bias found in the Base indicate that the application of the soil Nr emission schemes can effectively improve the simulation performance of MDA8 $O_3$."

In this study, we compare the simulated concentrations of $NO_2$, HONO, $O_3$, and nitrate in the atmosphere with and without the implementation of the soil Nr emission scheme against the observations. The evaluations of these air pollutions were used to validate the accuracy of the soil Nr emissions scheme incorporated into the UI-WRF-Chem. Specifically, by comparing the HONO concentrations simulated by the Default and Base, we aim to identify the accuracy of the SHONO scheme adopted in the model along with other four potential sources of HONO (i.e., traffic emissions, $NO_2$ heterogeneous reactions on ground and aerosol surfaces, and inorganic nitrate photolysis in the atmosphere). From Figure 2 to 4, the results show that the simulations with the implementation of BDISNP scheme are in better agreement with TROPOMI $NO_2$ VCD, observed surface HONO, $O_3$, and nitrate than the Default. Therefore, the soil Nr emission scheme has the fidelity needed to study the implication of soil Nr emissions on air quality in North China.

We modified the discussion in the revised version as follows:

Page 17-18 Lines 357-359: "Nevertheless, the improved simulation performance of $NO_2$ VCD, surface HONO, MDA8 $O_3$, and nitrate concentrations compared to the Default illustrates the credibility of the results obtained from the Base simulation."

*4. Emission Reduction Scenarios:*

*The manuscript discusses temperature responses to anthropogenic NOx emission changes. However, the scenarios focusing solely on NOx reduction may not reflect real-world conditions, as NOx is often co-emitted with other pollutants like SO2 during activities such as coal combustion. Therefore, the conclusions drawn from the current scenario setups might be skewed. I recommend including scenarios that consider reductions in emissions from co-emitted species to more accurately assess their collective impact on temperature.*

We admit that real conditions involve complex emission reduction scenarios that $NO_x$ is often co-emitted with other air pollutants, such as $SO_2$ and primary $PM_{2.5}$, particularly in coal combustion activities. Thus, the emission reduction scenarios conducted in our manuscript, which only focus on anthropogenic $NO_x$ emission reduction, may not adequately describe the multiple co-emission characteristics of air pollutants and their overall impact on temperatures. Notably, we focus on reducing anthropogenic $NO_x$ emissions mainly because the main objective of this study is to investigate the interaction and relative importance between anthropogenic and natural emissions of nitrogen-containing pollutants. We also believe that as anthropogenic emissions decrease to a certain extent, the impacts of natural emissions on air pollution and climate change mitigation become more important.

Future emission reduction scenarios are more complex, especially in terms of mitigating $O_3$ pollution, a certain emission reduction proportions of anthropogenic VOC and $NO_x$ should be maintained. The optimal reduction proportions can vary across different regions due to different $O_3$ formation sensitivity regimes, ranging from 1:1 to 4:1 (Guo et al., 2022; Ren et al., 2022; Wang et al., 2019). Therefore, to accept your point, and consider that future emission reduction scenarios should focus on the co-control of multiple air pollutants and greenhouse gas reductions (Cheng et al., 2021). we conducted additional anthropogenic emission reduction scenario experiments to reduce $SO_2$, $NO_x$, primary $PM_{2.5}$, VOCs, and CO emissions by 20%, 40%, 60%, 80%, and 100%, respectively, and evaluated the impact of soil Nr emissions on air temperature change under different anthropogenic emission reduction scenarios. The relevant discussions have been added in the revised version as follows:

Page 24, Lines 501-506: "Under the background of climate change, future emission reduction scenarios should focus on the co-control of multiple air pollutants and greenhouse gas reductions. Therefore, we conduct multi-pollutant co-control reduction scenarios, taking into account the $SO_2$, $NO_x$, primary $PM_{2.5}$, VOCs, and CO emissions reduced by 20%, 40%, 60%, 80%, and 100%, respectively, to investigate the impact of soil Nr emissions on air temperature change under different anthropogenic reduction scenarios (Table 1)."

Page 24-25, Lines 508-529: "Figure 8 shows that incorporating soil Nr emissions results in a slower rate of T2 increase compared to scenarios without soil Nr emissions, especially when multi-pollutant emissions are reduced to more than a half, and this phenomenon is consistent across all study regions. In the FWP region, when

anthropogenic emissions are eliminated, T2 increases by 0.073 °C in the presence of soil Nr emissions, compared to 0.095 °C in the absence of soil Nr emissions. In the BTH region, which has relatively high anthropogenic emissions, reducing multi-pollutant emissions by the same proportion could result in relatively greater warming, and T2 increases by 0.098 °C in the presence of soil Nr emissions, compared to 0.14 °C in the absence of soil Nr emissions when anthropogenic emissions are excluded. This is attributed to the effective radiative forcing (ERF) associated with the cooling effects of primary pollutants (e.g. $SO_2$, $NO_x$) and secondary inorganic aerosols (SIA), and positive ERF associated with the warming effects of CO and VOCs (high confidence) (Bellouin et al., 2020; Liao and Xie, 2021). Decreases in primary pollutants emissions and SIA concentrations could weaken the cooling effect and potentially accelerate warming to some extent, and the decrease in CO and VOCs emissions may still lead to temperature rise in a short-term. However, the soil Nr emissions could contribute to a certain background concentration of aerosol, partially offsetting the temperature rise caused by declining anthropogenic emissions of primary pollutants and greenhouse gas (Figure S8)."

***Specific comment***

*Line 237: monthly total? I suggest clarifying which month here.*

Thanks for the suggestion. We have revised it to "the monthly total $SNO_x$ are 18.7 Gg N mon$^{-1}$ in July". (Page 13, Line 268)

*Line 284-286: The authors attributed the positive biases to the same reasons documented by literature without mentioning more details. I assume literature uses similar settings in the default scenario. Do they have a similar magnitude of biases with the default or base scenario?*

Thanks for the suggestion. We have added more discussion in the revised version as follows:

Page 16-17 Lines 327-337: "Previous studies showed that the NMB of simulated $O_3$ concentrations were within ±30% for nearly 80% of the cases collected from air quality model studies (Yang and Zhao, 2023). These discrepancies may arise from simplifications of complex chemical mechanisms and physical processes, such as dry deposition and vertical mixing (Akimoto et al., 2019; Travis and Jacob, 2019). The uncertainties of input data, including emission inventories, meteorological fields, and other parameters, may also contribute to these discrepancies (Sun et al., 2019; Ye et al., 2022), suggesting a potential systematic $O_3$ bias in air quality models. Therefore, the increased spatial correlation and reasonable bias found in the Base indicate that the application of the soil Nr emission schemes can effectively improve the simulation performance of MDA8 $O_3$."

*Line 341: Any reasons given for the different conclusions with existing studies?*

Thanks for the suggestion. We have added the discussions in the revised version as follows:

Page 19-20 Lines 395-403: "The discrepancy between our findings and those of other studies regarding the impact of $SNO_x$ on ·OH levels could be attributed to the abundance of ambient $NH_3$ in China during summer, where soil emissions may lead to a significant increase in nitrate, and the increased aerosols can affect the concentrations of ·OH through photochemical reactions (Wang et al., 2011; Xu et al., 2022). Additionally, after taking into account the $SNO_x$ in the model, the environment may shift to a relatively $NO_x$-saturated regime, thus the termination reaction for $O_3$ production could be $NO_2$ and ·OH to generate $HNO_3$ (Chen et al., 2022; Wang et al., 2023b)."

*Conclusion: please clarify the contributions from soil Nr are not annual mean but for a specific month here.*

Thanks for the reminder. We have clarified the specific month in the Conclusion in the revised version as follows:

Page 26 Lines 541-543: "The contribution of soil Nr emissions in July to monthly average $NO_2$ and HONO are 38.4% and 40.3% in the BTH, and 33.9% and 40.1% in the FWP region, respectively"

*Figure 5: the statistical results are not easy to see. Suggest using an alternative color for the digits. Please also clarify the period used for the plotting in the caption.*

We have moved the statistics to the right corner of each panel and revised the caption in Figure 5. And also clarified the period in the caption of other figures in the revised manuscript and supplement.

*Figure 7: the legends of bars/lines are missing.*

We have added the legends of bars/lines in Figure 7.

Reference:

Akimoto, H., Nagashima, T., Li, J., Fu, J. S., Ji, D., Tan, J., and Wang, Z.: Comparison of surface ozone simulation among selected regional models in MICS-Asia III – effects of chemistry and vertical transport for the causes of difference, Atmos. Chem. Phys., 19, 603-615, 10.5194/acp-19-603-2019, 2019.

Bellouin, N., Quaas, J., Gryspeerdt, E., Kinne, S., Stier, P., Watson-Parris, D., Boucher, O., Carslaw, K. S., Christensen, M., and Daniau, A. L.: Bounding global aerosol radiative forcing of climate change, Rev. Geophys., 58, e2019RG000660, 2020.

Chen, K., Wang, P., Zhao, H., Wang, P., Gao, A., Myllyvirta, L., and Zhang, H.: Summertime $O_3$ and related health risks in the north China plain: A modeling study using two anthropogenic emission inventories, Atmos. Environ., 246, 118087, 10.1016/j.atmosenv.2020.118087, 2021.

Chen, W., Guenther, A. B., Jia, S., Mao, J., Yan, F., Wang, X., and Shao, M.: Synergistic effects of biogenic volatile organic compounds and soil nitric oxide emissions on summertime ozone formation in China, Sci. Total Environ., 828, 154218, 10.1016/j.scitotenv.2022.154218, 2022.

Cheng, J., Tong, D., Zhang, Q., Liu, Y., Lei, Y., Yan, G., Yan, L., Yu, S., Cui, R. Y., Clarke, L., Geng, G., Zheng, B., Zhang, X., Davis, S. J., and He, K.: Pathways of China's $PM_{2.5}$ air quality 2015–2060 in the context of carbon neutrality, Natl. Sci. Rev., 8, 10.1093/nsr/nwab078, 2021.

Guo, J., Zhang, X., Gao, Y., Wang, Z., Zhang, M., Xue, W., Herrmann, H., Brasseur, G. P., Wang, T., and Wang, Z.: Evolution of Ozone Pollution in China: What Track Will It Follow?, Environ. Sci. Technol., 57, 109-117, 10.1021/acs.est.2c08205, 2022.

Huang, L., Fang, J., Liao, J., Yarwood, G., Chen, H., Wang, Y., and Li, L.: Insights into soil NO emissions and the contribution to surface ozone formation in China, Atmos. Chem. Phys., 23, 14919-14932, 10.5194/acp-23-14919-2023, 2023.

Le, T., Wang, Y., Liu, L., Yang, J., Yung, Y. L., Li, G., and Seinfeld, J. H.: Unexpected air pollution with marked emission reductions during the COVID-19 outbreak in China, Science, 369, 702-706, 10.1126/science.abb7431, 2020.

Liao, H. and Xie, P.: The roles of short-lived climate forcers in a changing climate, Adv. Clim. Change Res., 17, 685, 2021.

Liu, F., Page, A., Strode, S. A., Yoshida, Y., Choi, S., Zheng, B., Lamsal, L. N., Li, C., Krotkov, N. A., Eskes, H., van der A, R., Veefkind, P., Levelt, P. F., Hauser, O. P., and Joiner, J: Abrupt decline in tropospheric nitrogen dioxide over China after the outbreak of COVID-19, Sci. Adv., 6, eabc2992, 10.1126/sciadv.abc2992, 2020.

Liu, Y., Geng, G., Cheng, J., Liu, Y., Xiao, Q., Liu, L., Shi, Q., Tong, D., He, K., and Zhang, Q.: Drivers of Increasing Ozone during the Two Phases of Clean Air Actions in China 2013–2020, Environ. Sci. Technol., 57, 8954-8964, 10.1021/acs.est.3c00054, 2023.

Ren, J., Guo, F., and Xie, S.: Diagnosing ozone–$NO_x$–VOC sensitivity and revealing causes of ozone increases in China based on 2013–2021 satellite retrievals, Atmos. Chem. Phys., 22, 15035-15047, 10.5194/acp-22-15035-2022, 2022.

Shen, Y., Xiao, Z., Wang, Y., Xiao, W., Yao, L., and Zhou, C.: Impacts of Agricultural Soil $NO_x$ Emissions on $O_3$ Over Mainland China, J. Geophys. Res.: Atmos., 128, 10.1029/2022jd037986, 2023.

Sun, L., Xue, L., Wang, Y., Li, L., Lin, J., Ni, R., Yan, Y., Chen, L., Li, J., Zhang, Q., and Wang, W.: Impacts of meteorology and emissions on summertime surface ozone increases over central eastern China between 2003 and 2015, Atmos. Chem. Phys., 19, 1455-1469, 10.5194/acp-19-1455-2019, 2019.

Tan, W., Wang, H., Su, J., Sun, R., He, C., Lu, X., Lin, J., Xue, C., Wang, H., Liu, Y., Liu, L., Zhang, L., Wu, D., Mu, Y., and Fan, S.: Soil Emissions of Reactive Nitrogen Accelerate Summertime Surface Ozone Increases in the North China Plain, Environ. Sci. Technol., 57, 12782-12793, 10.1021/acs.est.3c01823, 2023.

Travis, K. R. and Jacob, D. J.: Systematic bias in evaluating chemical transport models with maximum daily 8 h average (MDA8) surface ozone for air quality applications: a case study with GEOS-Chem v9.02, Geosci. Model Dev., 12, 3641-3648, 10.5194/gmd-12-3641-2019, 2019.

van Geffen, J., Boersma, K. F., Eskes, H., Sneep, M., ter Linden, M., Zara, M., and Veefkind, J. P.: S5P

TROPOMI NO$_2$ slant column retrieval: method, stability, uncertainties and comparisons with OMI, Atmos. Meas. Tech., 13, 1315-1335, 10.5194/amt-13-1315-2020, 2020.

van Geffen, J. H. G. M., Eskes, H. J., Boersma, K. F., and Veefkind, J. P.: TROPOMI ATBD of the total and tropospheric NO$_2$ data products, Report S5P-KNMI-L2-0005-RP, version 2.2.0, 2021-06-16, KNMI, De Bilt, The Netherlands, http://www.tropomi.eu/data-products/nitrogen-dioxide/ (last access: 7 March 2022), 2021.

Wang, H., Huang, C., Tao, W., Gao, Y., Wang, S., Jing, S., Wang, W., Yan, R., Wang, Q., An, J., Tian, J., Hu, Q., Lou, S., Pöschl, U., Cheng, Y., and Su, H.: Seasonality and reduced nitric oxide titration dominated ozone increase during COVID-19 lockdown in eastern China, npj Clim. Atmos. Sci., 5, 10.1038/s41612-022-00249-3, 2022a.

Wang, N., Lyu, X., Deng, X., Huang, X., Jiang, F., and Ding, A.: Aggravating O$_3$ pollution due to NO$_x$ emission control in eastern China, Sci. Total Environ., 677, 732-744, 10.1016/j.scitotenv.2019.04.388, 2019.

Wang, S., Xing, J., Jang, C., Zhu, Y., Fu, J. S., and Hao, J.: Impact Assessment of Ammonia Emissions on Inorganic Aerosols in East China Using Response Surface Modeling Technique, Environ. Sci. Technol., 45, 9293-9300, 10.1021/es2022347, 2011.

Wang, W., Parrish, D. D., Wang, S., Bao, F., Ni, R., Li, X., Yang, S., Wang, H., Cheng, Y., and Su, H.: Long-term trend of ozone pollution in China during 2014–2020: distinct seasonal and spatial characteristics and ozone sensitivity, Atmos. Chem. Phys., 22, 8935-8949, 10.5194/acp-22-8935-2022, 2022b.

Wang, W., Li, X., Cheng, Y., Parrish, D. D., Ni, R., Tan, Z., Liu, Y., Lu, S., Wu, Y., Chen, S., Lu, K., Hu, M., Zeng, L., Shao, M., Huang, C., Tian, X., Leung, K. M., Chen, L., Fan, M., Zhang, Q., Rohrer, F., Wahner, A., Pöschl, U., Su, H., and Zhang, Y.: Ozone pollution mitigation strategy informed by long-term trends of atmospheric oxidation capacity, Nat. Geosci., 17, 20-25, 10.1038/s41561-023-01334-9, 2023b.

Wang, Y., Fu, X., Wang, T., Ma, J., Gao, H., Wang, X., and Pu, W.: Large Contribution of Nitrous Acid to Soil-Emitted Reactive Oxidized Nitrogen and Its Effect on Air Quality, Environ. Sci. Technol., 57, 3516-3526, 10.1021/acs.est.2c07793, 2023c.

Xu, W., Zhao, Y., Wen, Z., Chang, Y., Pan, Y., Sun, Y., Ma, X., Sha, Z., Li, Z., Kang, J., Liu, L., Tang, A., Wang, K., Zhang, Y., Guo, Y., Zhang, L., Sheng, L., Zhang, X., Gu, B., Song, Y., Van Damme, M., Clarisse, L., Coheur, P.-F., Collett, J. L., Goulding, K., Zhang, F., He, K., and Liu, X.: Increasing importance of ammonia emission abatement in PM$_{2.5}$ pollution control, Sci. Bull., 67, 1745-1749, 10.1016/j.scib.2022.07.021, 2022.

Yang, J. and Zhao, Y.: Performance and application of air quality models on ozone simulation in China – A review, Atmos. Environ., 293, 119446, 10.1016/j.atmosenv.2022.119446, 2023.

Ye, X., Wang, X., and Zhang, L.: Diagnosing the Model Bias in Simulating Daily Surface Ozone Variability Using a Machine Learning Method: The Effects of Dry Deposition and Cloud Optical Depth, Environ. Sci. Technol., 56, 16665-16675, 10.1021/acs.est.2c05712, 2022.

Yienger, J. J. and Levy, H.: Empirical model of global soil-biogenic NO$_x$ emissions, J. Geophys. Res., 100, 11447–11464, 10.1029/95JD00370, 1995.

Yin, H., Lu, X., Sun, Y., Li, K., Gao, M., Zheng, B., and Liu, C.: Unprecedented decline in summertime surface ozone over eastern China in 2020 comparably attributable to anthropogenic emission reductions and meteorology, Environ. Res. Lett., 16, 124069, 10.1088/1748-9326/ac3e22, 2021.